# Analysis and Optimization Based on Factors Affecting the Spiral Climbing Locomotion of Snake-like Robot

Peng Zhang [1], Yong Zang [1,2], Ben Guan [1,2,*], Zhaolin Wu [1] and Zhiying Gao [1,2]

1   School of Mechanical Engineering, University of Science and Technology Beijing, Beijing 100083, China
2   Shunde Innovation School, University of Science and Technology Beijing, Foshan 528399, China
*   Correspondence: guanben@ustb.edu.cn; Tel.: +86-10-62334197

**Abstract:** The snake-like robot is a limbless bionic robot widely used in unstructured environments to perform tasks with substantial functional flexibility and environmental adaptability in complex environments. In this paper, the spiral climbing motion of a snake-like robot on the outer surface of a cylindrical object was studied based on the three-dimensional motion of a biological snake, and we carried out the analysis and optimization of the motion-influencing factors. First, the spiral climbing motion of the snake-like robot was implemented by the angle control method, and the target motion was studied and analyzed by combining numerical and environmental simulations. We integrated the influence of kinematics and dynamics factors on the spiral climbing motion. Based on this, we established a multi-objective optimization function that utilized the influence factors to optimize the joint module. In addition, through dynamics simulation analysis, the change of the general clamping force of the snake-like robot's spiral climbing motion was transformed into the analysis of the contact force between the joint module and the cylinder. On the basis of the results, the effect of the control strategy adopted in this paper on the motion and change rule of the spiral climbing motion was analyzed. This paper presents the analysis of the spiral climbing motion, which is of great theoretical significance and engineering value for the realization of the three-dimensional motion of the snake-like robot.

**Keywords:** snake-like robot; spiral climbing; influencing factors; optimization design; locomotion analysis

## 1. Introduction

The snake-like robot is bionic, with two free ends and multiple joints in tandem. Compared with the traditional legged mobile robots with a single or a few degrees of freedom, the snake-like robot has unique advantages and functional characteristics of multiple redundant degrees. It is, accordingly, extensively used in many fields, such as disaster rescue, work-state inspection, aerospace exploration, med-science research, and military reconnaissance [1–6].

Depending on the natural environment, snakes are capable of achieving a variety of locomotor gaits [7]. In particular, serpentine, rectilinear, and concertina locomotion are the most typical planar gaits. Moreover, these can accomplish many forward locomotor goals in planar environments. Three-dimensional gaits are developed to adapt to atypical environments and consist of sidewinding movements, intimidating movements when facing predators, and spiral climbing movements on tree trunks or the outer walls of pipes. Three-dimensional gaits are a complex gait of the snake that combines environmental factors and its characteristics with extreme functionality and adaptability. Among these, the spiral climbing motion extends the forward movement capabilities of the snake in planar motion to three-dimensional space, enriching the survival space of the snake in nature. Thus, studying the spiral climbing motion of snake-like robots is conducive to improving biomimicry and environmental interaction.

Generally, researchers have conducted studies related to the 3D motions of snake-like robots from different perspectives.

Hatton proposed a tread-based model for sidewinding. The new interpretation of the gait further admits a symmetry-based model reduction, and the behavior of an eclipse between the snake-like robot and the sloped surface in rolling contact was comprehensively analyzed [8]. Gong proposed a new and geometrically intuitive method to study the snake-like robot's steering strategy and turn rates of sidewinding gait by tapering the core cylinder into a cone [9]. Qi combined the hyperbolic curve with the helical curve in space to propose a new motion of helical wave propagation. By changing the hyperbolic function's parameters, the composite curve's stable propagation was achieved [10]. Yaqub developed a spiral curve gait whose joint angles are calculated by a Bellows model based on the curvature and torsion of the backbone curve, in which the rolling motion of the snake-like robot adapting to the variable diameter of the climbing pole was studied [11]. Rollinson found that the periodic gait of the snake-like robot was characterized by symmetry in form. They used the virtual chassis method to separate the different gaits of the snake-like robot into internal motion and external motion [12]. Zhou proposed a spring-like type of robot climbing pipe gait to solve the problem that the existing climbing pipe has a high demand of continuity of the pipe, employing variable curvature and variable torque discretization [13].

We divide the method that researchers in the current study on the 3D gait of snake-like robots into two main categories: curve approximation method [14] and curve discretization [13]. The former achieves the target curve's fitting by controlling the form of the snake-like robot. At the same time, the latter discretizes the ideal curve into nodes and controls the nodes' higher-order physical parameters to realize the snake-like robot's control. Although researchers can adopt both curve analysis and higher-order parameters to achieve 3D motion gait control, the correlation and influence between motion gait and the spatial shapes of snake-like robots still need to be improved in the current research.

Therefore, we studied the spiral climbing motion of the snake-like robot in this paper. The main contents are as follows: in Section 2, we realize the control of the spiral climbing motion of the snake-like robot, and the influencing factors affecting the motion state are obtained by combining kinematic and dynamic analysis methods. In Section 3, the cost function of each factor is established. Based on this, we create a multi-objective optimization function towards the robot's joint module, and thus, we complete the optimization analysis of the joint module design parameters. In Section 4, this paper shows a prototype of the snake-like robot, and the spiral climbing effect of the robot is verified and analyzed by simulation experiments. In the last chapter, we present the conclusion and discussion.

## 2. Analysis of Locomotion Patterns

There are two main types of spiral climbing motions of snakes [7,15], which are referred to as "folds and stretches" and "spirals and laterals" according to the different motion features. As shown in Figure 1A, the former type of movement is "head extension—head wrapping—tail contraction—tail wrapping". The snake achieves its entire spiral climbing through periodic changes of partial movements. In the latter case, as shown in Figure 1B, only part of the snake's body forms a spiral, while the other part forms a bend in the cylindrical surface and moves upward laterally, and this bend can alternate from side to side frequently.

The spiral climbing motion in nature is undoubtedly the most logical and efficient way to move [16], but it is too difficult to achieve for snake robots. The fundamental reason is that the joint module of the robots is less miniature and quantitative than a snake bone. Thus, researchers have modified this technically limited motion into a spiral rolling system.

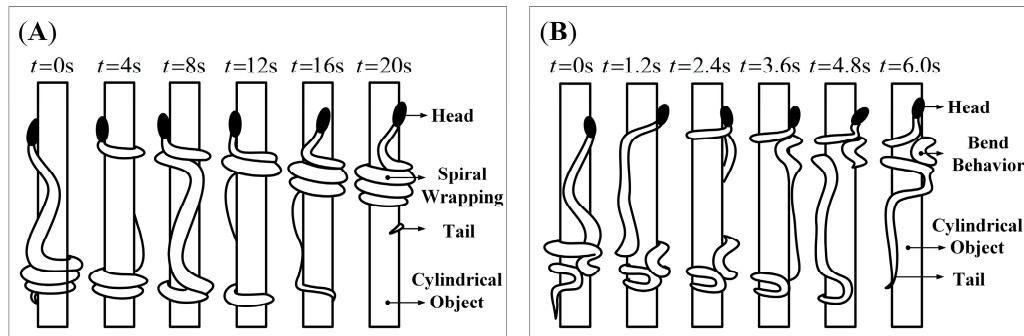

**Figure 1.** Different types of spiral climbing movement by a snake: (**A**) "Folds and Stretches"—wrapping helically and moving forward; and (**B**) "Spirals and Laterals"—Gripping during lateral undulation.

### 2.1. Method for Locomotion

In the current research, there are three main types of motion control methods for snake-like robots: the continuum method, the central pattern generator (CPG) method, and the function control method.

The Frenet–Serret expansion equation is the crucial point to the continuum model. Yamada first proposed the continuous model of active cord mechanism (ACM) and specifically classified it into the planar, Frenet–Serret, Bellows, and complete models according to the characteristics, and provided a method to deal with ACM's 3D shape [17]. Kamegawa realized cylinder locomotion with helical form by a snake-like robot using Yamada's ACM theory, and then the results were applied to a mechanical discrete snake-like robot model [18]. Qi developed a discrete control model for the joints of a snake-like robot using a continuous curve model and proposed a novel obstacle avoidance strategy for the robot wraps around the outside of a pipe [10,18]. Zhou's general parameter-based method discretized the spatial curve into the model with variable curvature and torque and applied the model of motion for gait by dividing functional areas to complete climbing over a stepped shaft and a discontinuous pipe [13]. Yaqub solved and calculated the joint angle of the snake-like robot based on the Bellows model utilizing a curvature integration algorithm [11]. Manzoor proposed a new algorithm for generating different rhythmic motions based on CPG, such as serpentine, sidewinding, two-step concertina, and four-step concertina [19]. Chirikjian presented an efficient kinematic modeling method for a snake-like robot based on the backbone curve, which reduces the inverse kinematics to the time-varying relative to the reference coordinate system of the backbone curve to describe the macroscopic geometric properties of the snake-like robot [20]. Lipkin described two categories of differential gaits: differentiable gaits; and segmented differentiable gaits. Based on the high redundancy characteristics of the snake-like robot, they established the function of the differentiable gaits and verified the feasibility of the gaits through experiments [21]. Choset proposed the CMU control model, which numbers the orthogonally connected joint modules of snake-like robots and outputs sinusoidal function control waves to the odd and even joint modules, respectively. Therefore the CMU control function is also known as a compound serpenoid curve [8,9,12,22]. Through the study of the spiral climbing gait of the snake-like robot, Sun proposed the angle control function model based on isometric spiral trajectory, which established the functional relationship between the form spiral trajectory angle and the joint angle. They analyzed the mechanical equilibrium performance of the snake-like robot's climbing gait [23]. Based on the serpentine curve proposed by Hirose, Wei compounded the motion model with the cylindrical helix equation to present a new simplified control function model, which established the relationship between the joint angle and time for the robot, and verified that it could accomplish a variety of motion gaits through experiments [24].

This paper focuses on the implementation and analysis of the spiral climbing motion developed by the snake-like robot, which requires both accurate output and control of the

joint modules. In addition, it is significant that a robust motion control method is the key to the spiral climbing state of the snake robot. Consequently, this paper is unique due to the comprehensive analysis and optimization of the influencing factors on the robot's spiral climbing motion.

### 2.2. Control of Spiral Climbing Locomotion

The flexibility of the snake is derived from its multi-joint structure of physiological characteristics. Hence in this paper, we simplified the snake to a multi-link mechanism, as shown in Figure 2, assuming that no lateral sliding occurs during the spiral climbing motion [18,24–26].

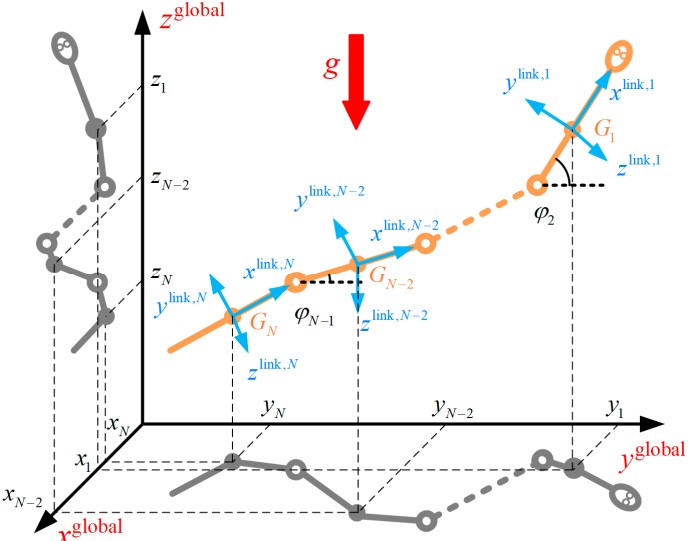

**Figure 2.** The Multi-link model of the snake-like robot.

Hirose first introduced the serpentine curve, which allows the serpentine gait of the snake-like robot by controlling the change of joint angle, and its control equation is described as:

$$\varphi_i = A\sin(\omega t + (i-1)\beta) + \lambda \tag{1}$$

The serpentine motion is a planar gait, but the spiral climbing motion is a 3D gait, for which a simplified cylindrical helix equation is introduced. As a result, we obtain the expression by compounding it with the angle control equation of the serpentine motion as follows:

$$\begin{cases} x = A\sin(\omega t) \\ y = A\cos(\omega t) \\ z = t \end{cases} \tag{2}$$

$$\varphi(i,t) = \begin{cases} A_{\text{even}}\sin(\omega t + (i-1)\beta_{\text{even}}) + \lambda_{\text{even}} \\ A_{\text{odd}}\sin(\omega t + (i-1)\beta_{\text{odd}}) + \lambda_{\text{odd}} \end{cases} \tag{3}$$

According to the connection order, we sort the joints of the snake-like robot into two categories, odd and even. $A$ is the amplitude, which controls the rotational direction of the snake-like robot in the spiral climbing motion. $\omega$ is the frequency controlling the execution rate of the actuator (time part). $\lambda$ is the compensation angle relative to the spiral centerline, and it controls the motion direction of the snake-like robot (space part). And $\beta$ is the phase difference. According to the experiment, it is learned that the snake-like robot moves in rectilinear or inward climbing gait when $\beta = 0$, while spiral climbing gait when $\beta \neq 0$.

This paper presents the control parameters $A_{\text{odd}} = A_{\text{even}} = A$, $\beta_{\text{odd}} = \beta_{\text{even}} = \beta \neq [0 \quad \pi/2]$, and $\lambda_{\text{odd}} = \lambda_{\text{even}} = \lambda$. The joint angle control function for the spiral climbing motion of the snake robot is as follows:

$$\varphi(i, t) = A \sin(\omega t + \beta i) + \lambda \tag{4}$$

Figure 3 shows that we can obtain the change in the snake-like robot's spiral climbing gait by adjusting the control parameters $A$, $\beta$, and $\lambda$, respectively. We can learn that the parameter $A$ affects the pitch of the spiral climbing gait, and the more significant $A$ is, the more extensive the output range of the joint angle becomes. Accordingly, the complete spatial pattern of the robot shows a more twisted state at this time. Parameter $\beta$ determines the radius of the robot's spiral, and the radius decreases with increasing $\beta$. In this case, the variation of the adjacent joint angle becomes more prominent, which in turn decreases the pitch of the spiral climbing gait. Thus, the robot shows a compressed state as a result. Parameter $\lambda$ has no significant effect on the motion itself, except that it changes the direction of the robot's spiral at the global level. Thereby the effect of $\lambda$ is not considered in the subsequent studies.

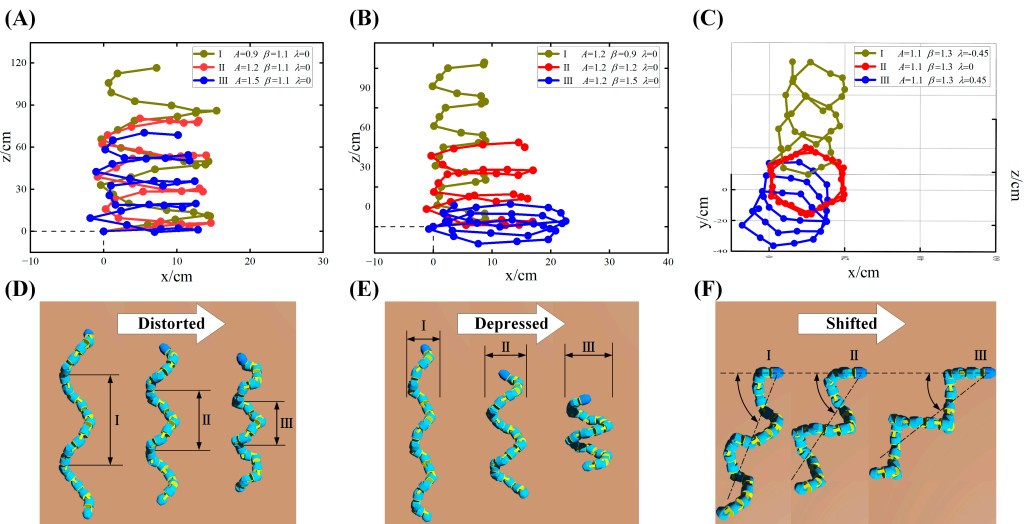

**Figure 3.** Different spiral climbing gaits when changing the parameters: (**A**,**D**) the effect of $A$ on spiral climbing gait; (**B**,**E**) the effect of $\beta$ on spiral climbing gait; (**C**,**F**) the effect of $\lambda$ on spiral climbing gait.

*2.3. Influencing Factors*

2.3.1. Radius of Spiral Climbing Gaits

From the analysis of Section 2.2, we find that both $A$ and $\beta$ have a determinative effect on the spiral climbing gait, as reflected in that by changing a single parameter, the radius and pitch of the form spiral of the snake-like robot will change. As shown in Figure 4, there exists any one joint module in contact with the surface of the cylindrical object when the snake robot wraps its body around the surface of the cylinder. We describe the two ends of the module are described as $\mathbf{p}_i = \begin{bmatrix} x_i & y_i & z_i \end{bmatrix}^{\text{T}}$ and $\mathbf{p}_{i+1} = \begin{bmatrix} x_{i+1} & y_{i+1} & z_{i+1} \end{bmatrix}^{\text{T}}$, respectively, using the improved D-H parameter method, at which the distance between the joint module and the centerline of the cylindrical object can be expressed as:

$$d_i^{i+1} = (y_{i+1} x_i - y_i x_{i+1}) \left( (x_{i+1} - x_i)^2 + (y_{i+1} - y_i)^2 \right)^{-1/2} \tag{5}$$

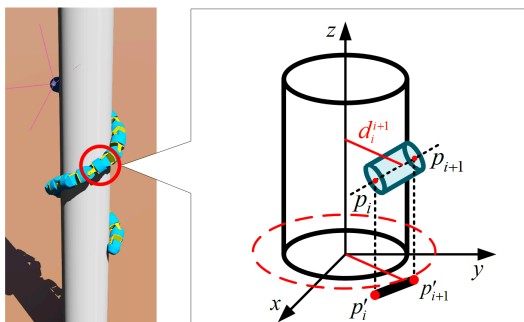

**Figure 4.** The single joint module in contact with the surface of a cylinder.

Ideally, the midpoint of the joint module should coincide with the generatrix of the cylinder, as described by the following equation:

$$\frac{x - x_0}{q_1} = \frac{y - y_0}{q_2} = \frac{z - z_0}{q_3} \tag{6}$$

Consequently, we equate the distance between the joint and the centerline of the cylinder to the radius of the form spiral of the robot's gait:

$$
R_d = d_i^{i+1}
$$
$$
= \frac{\begin{vmatrix} x_i - x_0 & y_i - y_0 & z_i - z_0 \\ x_{i+1} - x_i & y_{i+1} - y_i & z_{i+1} - z_i \\ q_1 & q_2 & q_3 \end{vmatrix}}{\left( \begin{vmatrix} y_{i+1} - y_i & z_{i+1} - z_i \\ q_2 & q_3 \end{vmatrix}^2 + \begin{vmatrix} z_i - z_0 & x_i - x_0 \\ q_3 & q_1 \end{vmatrix}^2 + \begin{vmatrix} x_i - x_0 & y_i - y_0 \\ q_1 & q_2 \end{vmatrix}^2 \right)} \tag{7}
$$

The spatial spiral radius of the snake-like robot is associated with six parameters, including $x_0$, $y_0$, $z_0$, $q_1$, $q_2$, and $q_3$. In order to establish the relationship among these parameters, this paper uses the particle swarm optimization algorithm (PSO) to optimize it. We use 31 sets of data from 30 joint modules as samples, with the minimum mean square deviation of $R_d$ as the optimization objective, set the number of examples of optimization $n = 31$, the maximum number of iterations of optimization $t_{ger} = 5000$, and the learning factor $c_1 = c_2 = 2.05$. Since there is uniqueness in the parameters obtained after optimization, this paper randomly selects one set of solutions, and Table 1 shows the data obtained by optimization:

**Table 1.** The radius of spiral climbing gaits by PSO.

|            | 1    | 2     | 3    | 4     | 5    | 6    | 7    | 8     | 9    | 10   |
|------------|------|-------|------|-------|------|------|------|-------|------|------|
| $A$        | 0.40 | 0.40  | 0.50 | 0.50  | 0.60 | 0.66 | 0.80 | 0.80  | 1.00 | 1.00 |
| $\lambda$  | 1.00 | 1.30  | 0.90 | 1.35  | 1.30 | 0.70 | 1.00 | 1.20  | 1.20 | 1.35 |
| $R_d$/cm   | 5.19 | 17.27 | 4.53 | 21.86 | 9.78 | 3.34 | 7.40 | 12.49 | 6.46 | 8.78 |

The fitting equation between the control parameters and the spatial spiral radius obtained based on the PSO establishment is as follows:

$$R_d = \frac{c_1 + c_2 A + c_3 A^2 + c_4 \lambda}{1 + c_5 A + c_6 A^2 + c_7 \lambda + c_8 \lambda^2} \tag{8}$$

Table 2 shows the 8 optimal values of Equation (8).

**Table 2.** The optimal values of the fitting equation.

|  | $c_1$ | $c_2$ | $c_3$ | $c_4$ | $c_5$ | $c_6$ | $c_7$ | $c_8$ |
|---|---|---|---|---|---|---|---|---|
| Optimal value | $-0.0812$ | 2.2446 | $-0.5158$ | 0.0149 | 0.0187 | 0.0756 | $-1.2878$ | 0.4128 |

### 2.3.2. Contact Point

During the ideal spiral climbing motion, the contact mode between the joint module of a snake-like robot and the surface of a cylindrical object can be classified into two categories, central point contact and non-central point contact [25], as shown in Figure 5. Since the snake robot is a tandem multi-joint robot, the contact mode of the first joint determines the general contact mode. Hence, the contact between the joint modules and the cylindrical surface is usually non-center point contact in the practical motion which is highly random.

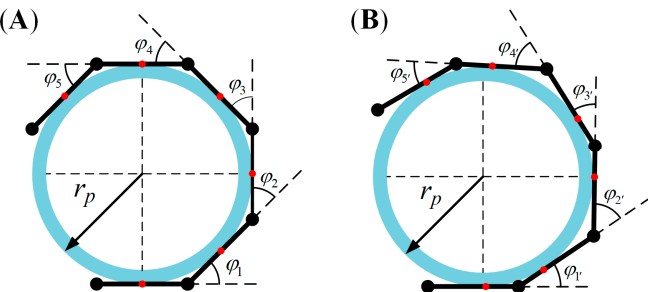

**Figure 5.** The contact models of a snake-like robot with the cylinder: (**A**) Central point contact model. (**B**) Non-central point contact model.

For further analysis of the contact circumstances, we simplify the adjacent joint modules into a shuttle-like structure with connected heads and tails, as shown in Figure 6.

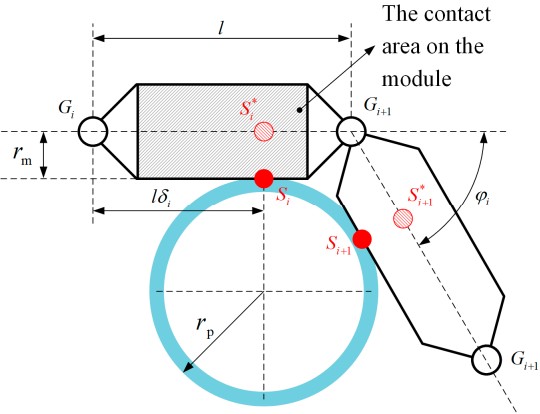

**Figure 6.** The contact geometry between the adjacent joint module and cylinder.

Taking $\delta_i = \overline{S^*G_i}/l \in [0,1]$ represents the coefficient of contact point position, and $\delta_i = 0.5$ means the central point contact. For non-central point contact, the larger $\delta_i$ is, the closer the contact point is to the backward joint. For t calculation, we determine that the coefficient of contact point position of the odd-joint module and the even-joint are to satisfy the requirement:

$$\delta_{\text{odd}} = 1 - \delta_{\text{even}} \tag{9}$$

The cylindrical coordinate system is established based on the cylinder, and then the contact point position matrix of the snake robot joint module is as follows:

$$\mathbf{S}_i = \left(r_p, \vartheta, z_{S_i}\right) \tag{10}$$

$$\vartheta = \sum_{i=0}^{n} \phi_i \tag{11}$$

$$\varphi_i = 2\arctan\left(\frac{(1 - \delta_i l \cos \varepsilon_i)}{r_p + r_m}\right) \tag{12}$$

$$z_{S_i} = (i - \delta_0 + \delta_i)l \sin \varepsilon_i \tag{13}$$

The end position of the joint module $G_i$ is represented in the cylindrical coordinate system as:

$$G_i = \left(r_{G_i}, \phi_{G_i}, z_{G_i}\right) \tag{14}$$

$$\phi_{G_i} = \sum_{i=0}^{n} \phi_i + \phi_i/2 \tag{15}$$

$$r_{G_i} = \sqrt{(r_m + r_p)^2 + ((1 - \delta_i)l \cos \varepsilon_i)^2} \tag{16}$$

$$z_{G_i} = (1 - c_0)l \cos \varepsilon_i \tag{17}$$

Figure 7 shows the projection of the two adjacent joint modules in contact with the cylinder in the $z = 0$ plane.

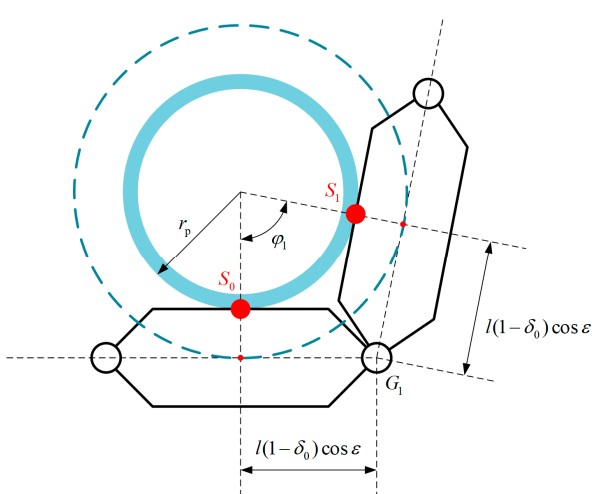

**Figure 7.** The contact between two adjacent modules and the cylinder.

The projected location of the contact point position on the joint centerline is expressed as follows:

$$\mathbf{S}_i^* = \mathbf{S}_i + \mathbf{r}_m \tag{18}$$

$$\mathbf{r}_m = (r_m, 0, 0) \tag{19}$$

$$\varphi_i = \pi - \arccos((\mathbf{a}_i \cdot \mathbf{b}_i)/\|\mathbf{a}_i\| \cdot \|\mathbf{b}_i\|) \tag{20}$$

$$\mathbf{a}_i = \mathbf{S}_i^* - \mathbf{G}_{i+1} \tag{21}$$

$$\mathbf{b}_i = \mathbf{G}_{i+1} - \mathbf{S}_{i+1}^* \tag{22}$$

$$\mathbf{a}_i \cdot \mathbf{b}_i = \|\mathbf{a}_i\| \cdot \|\mathbf{b}_i\| \cos(\pi - \varphi_i) \tag{23}$$

In the theoretical analysis, the spiral inclination angle of the robot's joint is $0 \leq \varepsilon < \pi/2$. Simultaneously, in this paper, we consider that $l$ and $r_p$, the length of the joint module of the snake-like robot and the radius of the cylinder, are satisfied to exist in a designing proportion. Due to the limitation of the mechanical structure, $\varphi_i \leq \varphi_{\max} < \pi/2$. We assume that $l \geq 2r_p$ ($\varepsilon = 0$), in this case there will be a problem that the joint angle of the snake-like

robot is all equal to $\pi/2$. Apparently, it does not meet the design requirements. Thus, we establish the design portion of the joints in this paper as follows:

$$l_{max} < 2(r_p + r_m) \cos \varepsilon \tag{24}$$

Figure 8 shows the parameters of the joint module. The angle of the joint module's shuttle part is the same as the range of the rotation angle of the joint, which is $[0, \pi/2]$. Therefore, we consider that the maximum rotation angle of the joint and the inclination angle of the shuttle part are identical, both of which are $\varphi_{max}$. The parameters of the joint module should meet the requirements as follows:

$$\max(r_m) = \tan \varphi_{max} l(1 - \delta_{max}) \tag{25}$$

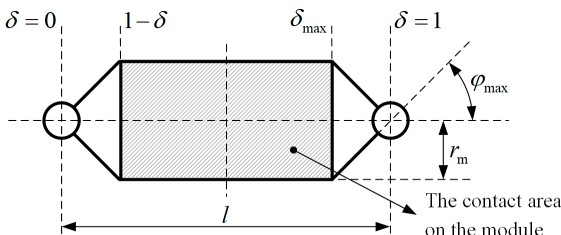

**Figure 8.** The parameters of a single joint module.

## 3. Analysis of Optimization

In the spiral climbing motion of the snake-like robot, influencing factors from kinematics and mechanics can disturb the motion itself. However, after the motion becomes stable, three critical factors are identified, including the number of joint modules of the snake-like robot, the forward velocity of the spiral climbing motion, and the output torque of the joints. As a result, this paper analyzes and optimizes the spiral climbing motion of the snake-like robot based on these three influencing factors.

### 3.1. Cost Function Based on Factors

### 3.1.1. The Number of Joint Modules

We learn that the number of joint modules impacts the state of the snake-like robot wrapping around the surface of the cylindrical object. Theoretically, the more joint modules connected by a snake-like robot, the larger the radius of the cylindrical object for the snake-like robot to wrap around, but the control cost will also increase. We can calculate the number of joint modules required to wrap spirally around the cylinder surface for one cycle by deriving the maximum angle between adjacent joint modules, and the cost of the spiral wrap based on the central point contact is expressed as follows:

$$cost_n = \frac{2\pi}{\max(\varphi_{odd})} = \frac{2\pi}{\max(\varphi_{even})} \tag{26}$$

In the case of non-central point contact, $\varphi_{odd}$ and $\varphi_{even}$ change with $c_{odd}$ and $c_{even}$. Therefore, the cost of spiral wrapping is as follows:

$$cost_n = 2 \times \frac{2\pi}{\varphi_{odd} + \varphi_{even}} \tag{27}$$

### 3.1.2. The Forward Velocity of Spiral Climbing

The forward velocity of the snake-like robot's spiral climbing motion is related to the radius $r_m$ of the joint module and the spiral inclination angle $\varepsilon$. As shown in Figure 9, the joint module with a larger radius has a faster forward velocity within the same spiral inclination angle $\varepsilon$.

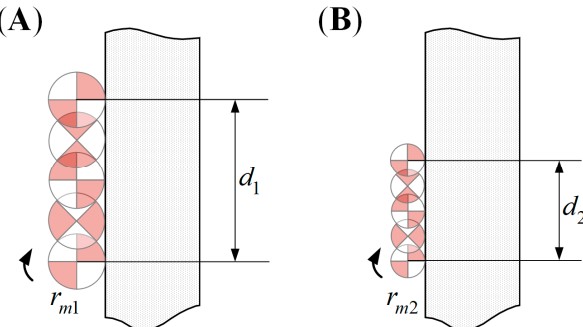

**Figure 9.** Comparison of modules with different radii: (**A**) joint module with radius of $r_{m1}$; and (**B**) joint module with radius of $r_{m2}$.

The forward velocity of the snake-like robot on the surface of the cylindrical object is the vertical component of the global velocity. Since the horizontal component of the global velocity causes the interaction between the robot itself and the cylinder, the cost function of the climbing velocity is as follows:

$$cost_v = \frac{\cos \varepsilon}{r_m} \tag{28}$$

### 3.1.3. The Output Torque of Joints

The output torque of the joint is the critical parameter to ensure the motion of snake-like robots. As shown in Figure 10, we simplify the joint module of the snake-like robot in contact with the surface of the cylindrical object to a single cantilever module for analysis. We also neglect the anterior module reaction force to the analyzed joint. Moreover, the reaction force to the joint module cancels off with the normal force and the horizontal component of the friction force. Meanwhile, considering that the analyzed joint module is an ideal linkage without thickness, the vertical component of the friction force cancels off with the gravitational force of the linkage, expressed as follows:

$$f_f = W = \mu f_N \tag{29}$$

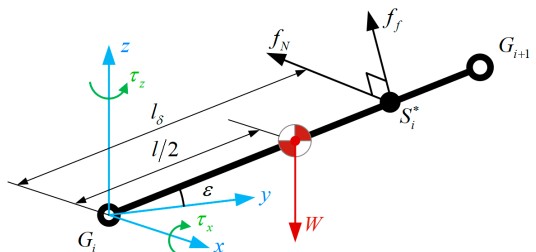

**Figure 10.** The forces of a single joint.

Since it is an ideal linkage without thickness, we can calculate the gravity of the joint module approximately as the volume of the module:

$$W = lr_m^2 \tag{30}$$

$$\tau_x = 0.5l \cos \varepsilon W - f_f l \delta_{\max} \tag{31}$$

$$\tau_z = f_N l \delta_{\max} \tag{32}$$

Thus, the cost of the torque to the joint module is as follows:

$$cost_\tau = \sqrt{\tau_x^2 + \tau_y^2} \tag{33}$$

### 3.2. Optimization Design

From Section 3.1, we analyze that different factors affect the spiral climbing motion from different perspectives. Therefore, this paper adopts a linear combination to consider the cost of these factors comprehensively. In order to take the weights and sensitivities of different factors into account, the weight coefficients $w = \begin{bmatrix} w_1 & w_2 & w_3 \end{bmatrix}$ and sensitivity factors $\sigma = \begin{bmatrix} \sigma_1 & \sigma_2 & \sigma_3 \end{bmatrix}$ are introduced, respectively. Hence the multi-objective optimization function of the spiral climbing motion is created by the cost function as follows:

$$cost = \sigma_1 w_1 cost_n + \sigma_2 w_2 cost_v + \sigma_3 w_3 cost_\tau \tag{34}$$

where Equation (34) uses the sensitivity factor to normalize the three influencing factors and make it possible to calculate them in the same order of magnitude, and it uses the sensitivity factor by sensitivity analysis.

$$\sigma_j = \frac{1}{Max_j} \tag{35}$$

There are restrictions on the parameters of the joints in the individual cost functions. $\delta_{max} = 0.5$ means that we choose the central contact model for calculation. Moreover, we define $l \in [0.2, 2]$ and $r_m \in [0.2, 2]$, both of which limit the joint module's size of the design to no larger than the cylinder's radius. Furthermore, $\varepsilon \in [0, \pi/2]$ requires that the snake-like robot can't be vertical relative to the ground. We calculate the sensitivity factor as $\sigma = \begin{bmatrix} 0.0819 & 0.3788 & 1.01 \end{bmatrix}$. Each weight in the multi-objective optimization function has to be positive, representing the importance of the corresponding influencing factor compared to the other two factors. Also, the three need to be content with $w_1 + w_2 + w_3 = 1$. Consequently, we can obtain the optimization parameters of a single joint module according to different cylinder and spiral climbing. Table 3 shows the optimization input parameters and the optimized outputs.

**Table 3.** The optimal design parameters of joint module when different cases.

| Symbol | Meaning | Case 1 | Case 2 | Case 3 | Case 4 |
|---|---|---|---|---|---|
| $r_p$ | Radius of Cylinder | 20 | 20 | 20 | 20 |
| $\mu$ | Friction Coefficient | 0.4 | 0.4 | 0.4 | 0.4 |
| $\varepsilon$ | Helical Pitch | 10 | 10 | 10 | 10 |
| $\varphi_{max}/^\circ$ | Maximum Rotation Angle | 75 | 75 | 75 | 75 |
| $\delta_i$ | Coefficient of Contact Point | 0.5 | 0.5 | 0.5 | 0.5 |
| $l_{min}/cm$ | Minimum Length of Module | 5 | 5 | 5 | 5 |
| $r_m\vert_{max}/cm$ | Maximum Length of Module | 5 | 5 | 5 | 5 |
| $w_1$ | Weight of $cost_n$ | 0.15 | 0.15 | 0.70 | 0.33 |
| $w_2$ | Weight of $cost_v$ | 0.15 | 0.70 | 0.15 | 0.33 |
| $w_3$ | Weight of $cost_\tau$ | 0.70 | 0.15 | 0.15 | 0.33 |
| $l/cm$ | Length of Module | 9.2326 | 11.2376 | 8.0152 | 12.1638 |
| $r_m/cm$ | Radius of Module | 4.2781 | 4.5013 | 3.0201 | 4.1032 |

According to Equation (34) and the constraints mentioned above, we get four cases based on weights for the snake-like robot's joint module parameters. The data in Table 3 clearly show the optimized length and radius of the joint module. Moreover, Due to the linear combination among the weights, the sensitivity factors, and the cost functions, different weights have a magnification effect on different influence factors. Therefore, these four cases form a comparison and reference to each other, and the optimization results are compelling. Figure 11 shows the snake-like robot whose joint module is designed based on the specific case.

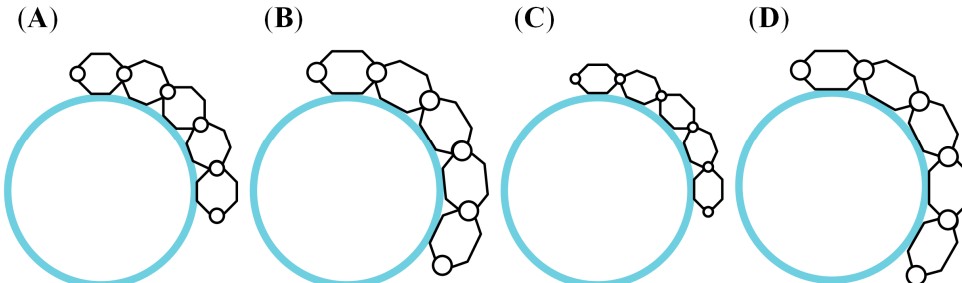

**Figure 11.** The snake-like robot whose joint module is designed based on the optimization cases: (**A**) case 1; (**B**) case 2; (**C**) case 3; and (**D**) case 4.

## 4. Simulation

In order to evaluate the performance of the snake-like robot with the optimized joint module. We design a snake-like robot with joint parameters $l = 12.16$ cm and $r_m = 4$ cm with a cylinder of $r_p = 20$ cm as the climbing target object, and the mechanical performance of this model is undertaken.

### 4.1. Modeling of the Snake-like Robot

Figure 12 illustrates the snake-like robot we designed, which has 20 joint modules connected orthogonally. A servo controls a single joint, and the rotation axes between two adjacent joints are perpendicular, making the snake-like robot have both ten pitch and ten yaw degrees of freedom, correspondingly.

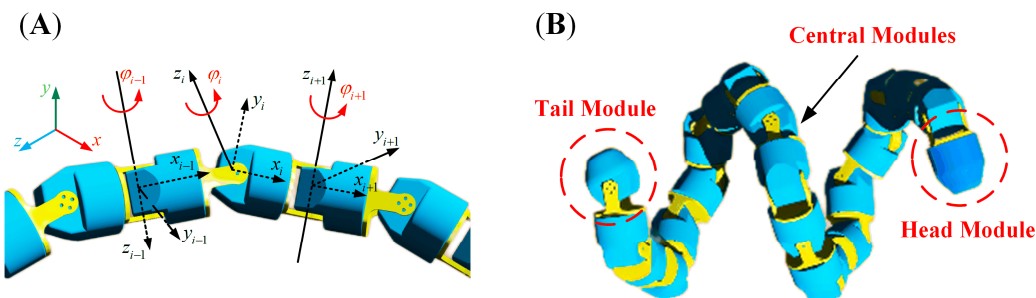

**Figure 12.** The snake-like robot with orthogonal joints: (**A**)the joint modules are connected orthogonally; and (**B**) the snake-like robot is modeled by 20 modules which can be divided into head, tail, and central modules.

We analyze the contact mode between the joint module and the cylinder in the process of climbing in Section 2.3. A reasonable contact state is an essential guarantee of the grip force required for the snake-like robot to wrap around the surface of the cylinder. Theoretically, the grip force is the sum of the frictional forces on all joint modules. However, the measurement and acquisition of the frictional forces are incredibly challenging to achieve in practice. Thus, this paper defines the contact force generated when the joint module is in contact with the cylinder as a collision. It can be obtained by dynamics simulation, which can transform the analysis of the grip force applied to the snake-like robot into that of contact force. Table 4 shows the setting parameters of the contact force constraint.

**Table 4.** The parameters of the contact force constraint.

| Parameter | Value | Parameter | Value |
|---|---|---|---|
| Stiffness ($N/\text{mm}$) | 2855.00 | Dynamic Friction Coeff. | 0.25 |
| Damping (($N \cdot \text{s})/\text{mm}$) | 0.57 | Static Friction Vel. (mm/s) | 0.10 |
| Exponent | 1.10 | Dynamic Friction Vel. (mm/s) | 10.00 |
| Penetration Depth (mm) | 0.10 | Coefficient of Restitution | 0.80 |
| Static Friction Coeff. | 0.30 | - | - |

The spiral climbing motion of the snake-like robot is shown in Figure 13.

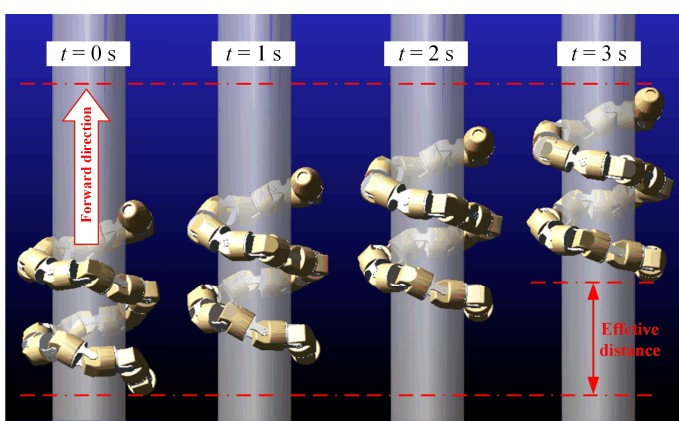

**Figure 13.** The Spiral climbing motion of the snake-like robot.

*4.2. Simulations and Results*

4.2.1. Vibration at Startup

The simulation indicates that in the moment of motion initiation, as the snake-like robot changes from the zero-moment state to the state of spiral climbing motion, the robot needs to break the original equilibrium state to another. Thereby, a shaking exists. Taking the head joint as an example, as shown in Figure 14, because of the contact between the head joint and the cylinder surface at the moment of the startup, the instantaneous contact force appears to change drastically, resulting in a vibration trend in the displacement of the head module. Furthermore, as the snake robot's posture gradually gets balanced, the joint module's displacement and contact force begin to change smoothly. Therefore, the first 0.5 s of the startup moment were ignored in the subsequent analysis to obtain a better analysis of the spiral climbing motion.

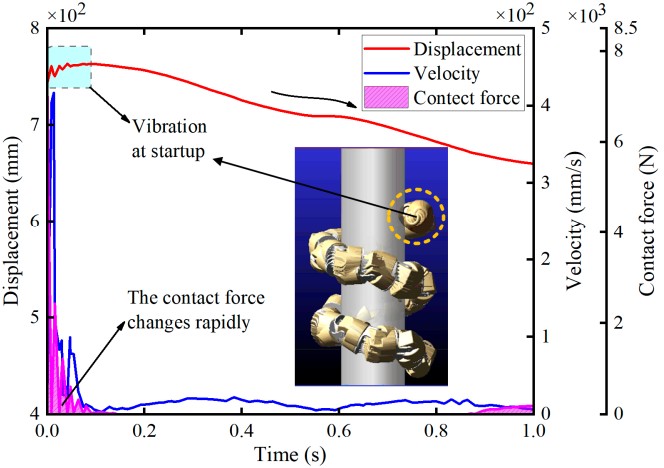

**Figure 14.** The vibration of the head module at startup.

### 4.2.2. Periodic Variation

Due to the tandem characteristics of the snake-like robot, there are interactions between adjacent joints. We select the central joint as the object of analysis in this paper.

The output torque and energy consumption of the 10th and 11th joints are analyzed as shown in Figure 15A. The output torque of the joints has the characteristic of periodic variation. However, there are multiple spikes in a single cycle, which is attributed to the gravity, friction, and inertia forces functioning simultaneously on the anterior and posterior joints at this time. This complicated force situation makes the combined force of the joint modules vary widely. Therefore, there are small fluctuations in the output within a single cycle, although the overall range of variation in output torque remains small. With the similarity between the joint module's instantaneous energy consumption and the variation of the output torque, that is, the cycle variation pattern and the energy consumption variation within a single cycle are not smooth curves.

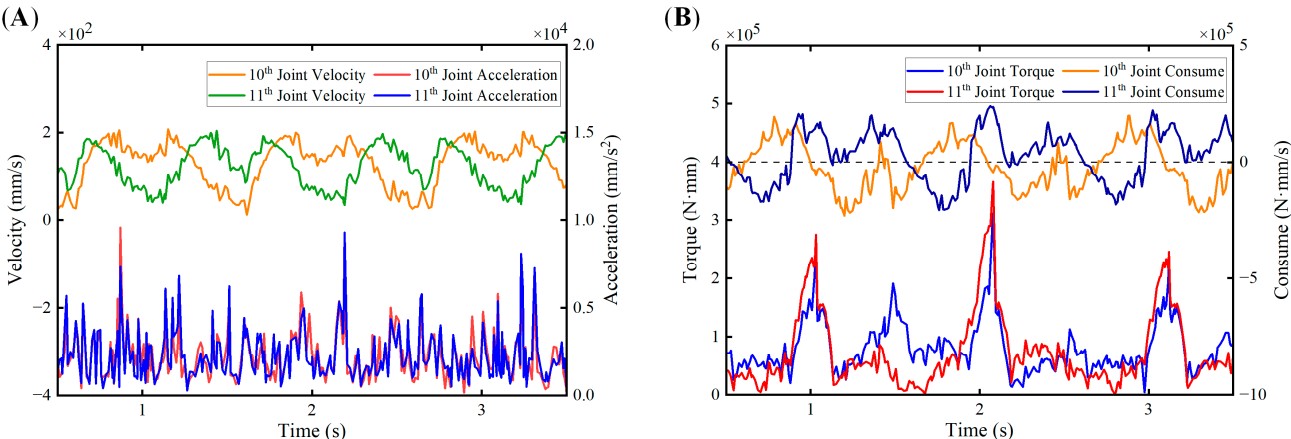

**Figure 15.** The mechanic characteristics of adjacent modules: (**A**) velocity and acceleration relative to time; and (**B**) joint torque and consumption relative to time.

Figure 15B shows the velocity and acceleration analysis of the 10th and 11th joints. The velocity variation of these joints maintains a good periodic pattern with a small scale. Due to the influence of the output torque fluctuation, acceleration is characteristic of periodic variation, but there are multiple spike bursts in a single cycle. However, the overall fluctuation of acceleration varies slightly, which verifies that the snake-like robot completes a smooth climbing motion on the cylinder after the changes from the starting vibration to the stable.

### 4.2.3. Contact Force

The snake-like robot continuously updates the state of contact with the cylinder surface through the change of joint output angle, as shown in Figure 16B, where the direction of the red arrow indicates the direction of the current joint contact force, and the length of the red line segment represents the magnitude. Figure 16A shows that the adjacent joints alternately have contact with the cylinder after being stable, and the contact force has a periodic characteristic. At the same time, multiple spikes of a sudden increase appear in the contact force within a single cycle but maintain a stable limit overall. The contact force in the two adjacent joints has a "delay" effect due to the phase difference between the output angle of the anterior and posterior joints.

**(A)**

**(B)**

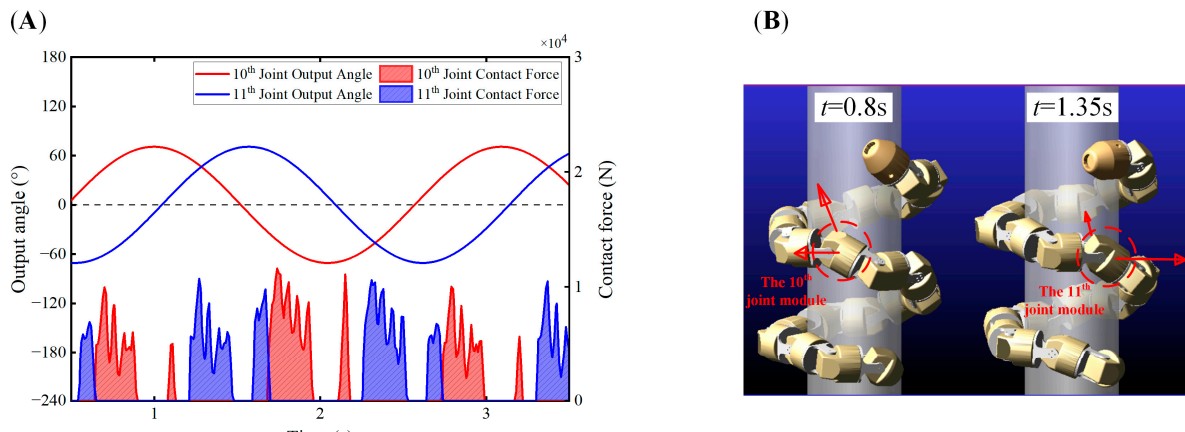

**Figure 16.** The output angle and contact force of two adjacent joints: (**A**) the output angle and contact force of joint modules; and (**B**) the contact force is applied to modules at different time points.

Figure 17 shows the pattern of output rotation angle and contact force variation of four successive adjacent central joints. In addition to the similar periodic regularity of contact force variation with two adjacent joints, not all the joints are in contact with the cylinder surface simultaneously. As illustrated in Figure 17B, only the 9th and 11th joints among the inspected joints are in contact with the cylinder surface when $t = 1.35s$. While $t = 1.95s$, only the 10th and 12th joints are in contact with the surface of the cylinder. Figure 17A clearly shows the delay effect between the contact cycles of the different joints. Different joints' contact forces will effectively cover the time axis to provide enough contact force for the snake-like robot to climb upwards stably within the same time frame.

**(A)**

**(B)**

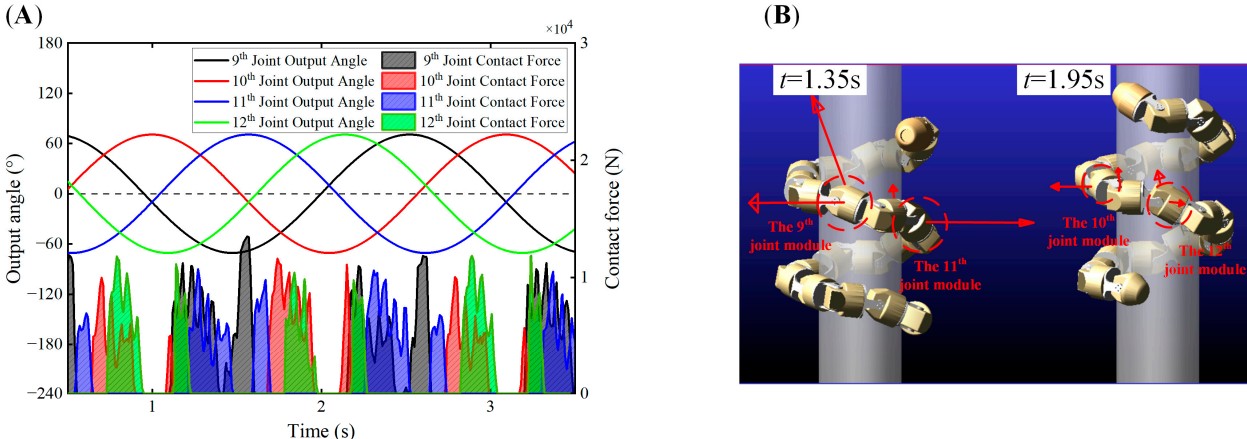

**Figure 17.** The output angle and contact force of four adjacent joints: (**A**) the output angle and contact force of joint modules; and (**B**) the contact force is applied to modules at different time points.

Figure 18 shows the joint output angles (JOA) and joint contact forces (JCF) of six successive adjacent joints. We consider the six successive adjacent joints of the snake-like robot we analyze as an integral unit, and that the output angles of different joints have a sinusoidal output pattern in one cycle. At any point under the cycle, the integral unit of the snake-like robot is in contact with the cylinder surface, which is necessary for generating sufficient grip force.

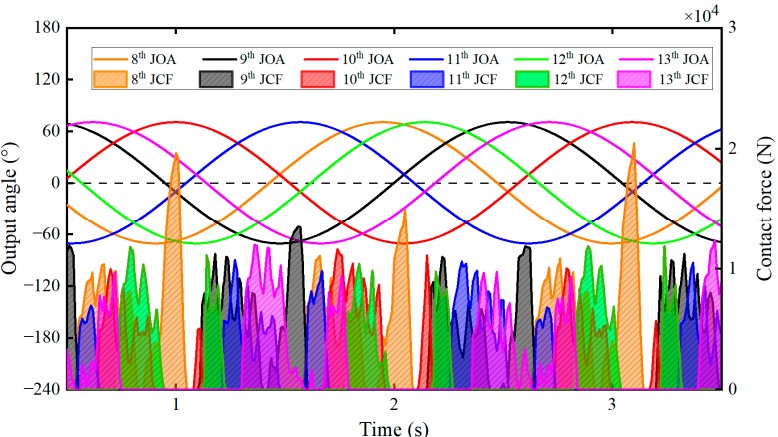

**Figure 18.** The output angle and contact force of six adjacent joints.

4.2.4. Non-Contact Zone

In the analysis mentioned above, we find that there are specific output angles with corresponding contact forces of joint modules at the same time. As shown in Figure 19, the joint module of the snake-like robot only makes contact with the surface of the cylinder in a particular range of angles during the spiral climbing motion, and the joint angles in the current state are not in contact with the surface of the cylinder in the range of $\pm 30°$.

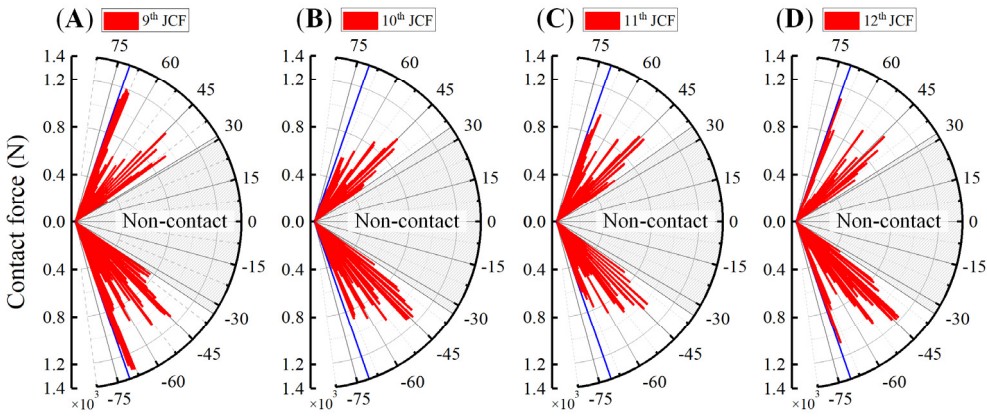

**Figure 19.** Non-contact zone: (**A**) JCF of the 9th module; (**B**) JCF of the 10th module; (**C**) JCF of the 11th module; and (**D**) JCF of the 12th module.

The non-contact zone is inevitable when the joint angle control method is applied to achieve spiral climbing motion. However, if the non-contact zone is too large, the contact between the joint modules and the cylindrical object will be reduced, which influences the climbing effect, while on the contrary, if the non-contact zone is too small, the joints will be in contact with the cylindrical object at all times, indirectly influencing the radius of the form spiral of the snake-like robot, which makes it incompatible for the robot to adapt to the cylinder of a larger diameter. Also, it still affects the climbing effect.

**5. Conclusions**

In this paper, we analyze the factors influencing the spiral climbing motion of a snake-like robot on the outer surface of a cylinder and its optimal design. Although many studies have been conducted on the spiral climbing motion, few have conducted analyses from the perspective of kinematics and dynamics-influencing factors.

First, the angle control method was used in this paper to realize the spiral climbing motion control of the snake-like robot. We analyzed the effect of control parameters $A$, $\beta$, and $\gamma$ on the motion utilizing MATLAB R2022a and WEBOTS 2021a. Moreover, we

establish a formulation of form spiral radius based on $A$ and $\beta$ to realize the directional control of the snake-like robot's spiral climbing motion. Then, a contact point location analysis determined the contact range of the joint module and the cylinder surface. A multi-objective optimization function with three principles was established based on the number of joint modules, the forward velocity of motion, and the output torque of the joint, to carry out a practical analysis and optimization of the design parameters of the joint module. In the final analysis, we transferred the grip force when the snake-like robot wrapped around the outer surface of the cylinder into the joint module's contact force. We clarified that the pattern of the grip force when taking the six consecutive adjacent joints as an integral unit shows that it is always in contact with the cylinder. Furthermore, the generation and influence of the non-contact zone were analyzed.

The work of the spiral climbing motion from kinematic and dynamic influences carried out in this paper is unique and instructive for analyzing the spatial motion of snake-like robots. In the subsequent research, we will focus on the spiral climbing motion of snake-like robots on the surface of cylinders with variable diameters.

**Author Contributions:** Methodology, formal analysis, investigation, writing—original draft preparation, P.Z.; supervision, validation, projection administration, Y.Z.; software, data curation, Z.W.; funding acquisition, writing—review and editing, B.G. and Z.G. All authors have read and agreed to the published version of the manuscript.

**Funding:** This research was funded by the Fundamental Research Funds for the Central Universities, grant number FRF-DF-20-13, and Foshan Science and Technology Innovation Fund, University of Science and Technology Beijing (USTB), China, grant number BK20BE007.

**Acknowledgments:** We would like to acknowledge the assistance of Wenhao Wang and Fan He, both of whom are the student at Shunde Innovation School, USTB (The former name is Shunde Graduate School, USTB). And we send our thanks to Xiaoya Zhao and Shanshan He for the software skills they have shared with us.

**Conflicts of Interest:** The authors declare no conflict of interest.

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
