# Peer review of "Analysis and Optimization Based on Factors Affecting the Spiral Climbing Locomotion of Snake-like Robot"

_electronics, doi:10.3390/electronics11234002_

Round 1
Reviewer 1 Report
This paper analyzes the spiral climbing motion for snake-like robots, mainly focusing on how to model and optimize the factors. There are some suggestions to enhance the manuscript further.
1. This is an interesting work, however too focused-on different design factors optimization discussions. When it comes to experimental demonstration, for example, section 4.2.2 didn’t show a clear comparison of what may happen without and with factor optimization, which could be hard to convince readers of the necessity of factor optimization.
2. There should be more citations relevant to this work; authors may want to add more, especially on building bridges from theoretical discussion to experimental demo and vice versa.

Reviewer 2 Report
Based on a careful analysis, I can formulate the following remarks:
1) The aim of this article, based on the author’s scrupulous investigations, is to solve the problem of the spiral climbing motion of a snake-like robot on the outer surface of a cylindrical object.
2) The topic represents in my opinion a relevant approach of the proposed theme, based on meticulous theoretical and investigations, correlated with relevant experimental results.
3) In comparison with other published material, the author’s contribution adds to the subject area a new approach/methodology, with several significant contributions.
In principle, their original approach consists in the followings:
- the spiral climbing motion of the snake-like robot is implemented by the angle control method, and the target motion is studied and analyzed by combining numerical simulation and environmental simulation;
- the influence of kinematics and dynamics factors on the spiral climbing motion is integrated, and a multi-objective optimization function based on the influence factors is established to optimize the joint module;
- through dynamics simulation analysis, the change of the general clamping force of the spiral climbing motion of the snake robot is transformed into the analysis of the contact force between the joint module and the cylinder;
- finally, the effect of the control strategy adopted in this paper on the motion and change rule of the spiral climbing motion is analyzed.
Based on the obtained results one can conclude that the proposed approach, i.e. the analysis of the spiral climbing motion, presents a great theoretical significance and engineering value for the realization of the three-dimensional motion of the snake-like robot.
4) The snake-like robot is a kind of bionic robot with two free ends and multiple joints in tandem. Compared with the traditional legged mobile robots with single or few degrees of freedom, the snake-like robot has the unique advantage and functional characteristics of multiple redundant degrees of freedom and consequently extensively used in many fields such as disaster rescue, work-state inspection, aerospace exploration, med-science research and military reconnaissance, etc.
Depending on the natural environment, snakes are capable of adaptively achieving a variety of locomotors gaits. In particular, serpentine, rectilinear, and concertina locomotion are the most typical planar gaits, and these are capable of accomplishing a wide range of forward locomotors goals in planar environments.
The developed three-dimensional gaits are adapted to atypical environments and consist of side-winding movements, intimidating movements when facing predators, and spiral climbing movements on tree trunks or the outer walls of pipes.
The three-dimensional gait represents a complex gait of the snake that combines environmental factors and its own characteristics, with extreme functionality and adaptability, among which, the spiral climbing motion extends the forward movement capabilities of the snake in planar motion to three-dimensional space, enriching the survival space of the snake in the nature.
Thus, the study of the spiral climbing motion of snake-like robots is conducive to improving the biomimicism and environmental interaction.
Generally, researchers have conducted studies related to the 3D motions of snake-like robot from different perspectives.
Researchers in the current study on the 3D gait of snake-like robots are divided into two main categories, which are:
· curve approximation method and
· curve discretization method.
The former achieves the fitting of the target curve by controlling the form of the snake-like robot, while the latter discretizes the ideal curve into nodes and controls the higher-order physical parameters of the nodes to realize the control of the snake-like robot.
Although both curve analysis and higher-order parameters can be used to achieve 3D motion gait control, the correlation and influence between motion gait and spatial shapes of snake-like robots are still lacking in the current research.
Based on these facts, the authors of this contribution analyzed the spiral climbing motion of the snake-like robot, such as:
- they completed the control of the spiral climbing motion of the snake-like robot;
- by combining kinematic and dynamic analysis methods obtained the influencing factors affecting the motion state;
- the cost function of each factor is established,
- and based on this, a multi-objective optimization function of the robot’s joint module is created, and the optimization analysis of the joint module design parameters is completed.
- Finally, by simulation experiments, the structural design of snake-like robot is completed, and the spiral climbing effect of the robot is verified and analyzed.
5) In my opinion, the presented conclusions are suitable related to their research results and prove that they reached the proposed goal.
6) The references in my opinion are very appropriate and their number underlines the scrupulosity of the authors.
7) One has to underline that the proposed approach represents a very efficient and promising step in the implementation and analysis of the spiral climbing motion developed by the snake-like robot, which requires both the accurate output and control of the joint modules. The developed approach represents also a robust motion control methodology, which in fact is the key to the spiral climbing state of the snake robot.
One has to remark the fact that both the graphical illustrations as well as the figures are very suggestive.
Based on the validation results, one can underline the efficiency of their proposed approach.
I encourage publishing in a new contribution his further results.
